# Studying Factors Affecting Success of Antimicrobial Resistance Interventions through the Lens of Experience: A Thematic Analysis

**DOI:** 10.3390/antibiotics11050639

**Published:** 2022-05-10

**Authors:** Tiscar Graells, Irene A. Lambraki, Melanie Cousins, Anaïs Léger, Kate Lillepold, Patrik J. G. Henriksson, Max Troell, Carolee A. Carson, Elizabeth Jane Parmley, Shannon E. Majowicz, Didier Wernli, Peter Søgaard Jørgensen

**Affiliations:** 1Global Economic Dynamics and the Biosphere, Royal Swedish Academy of Sciences, 114 18 Stockholm, Sweden; lillepoldkate@gmail.com; 2Stockholm Resilience Centre, Stockholm University, 106 91 Stockholm, Sweden; patrik.henriksson@beijer.kva.se (P.J.G.H.); max@beijer.kva.se (M.T.); 3School of Public Health Sciences, University of Waterloo, Waterloo, ON N2L 3G1, Canada; ilambrak@uwaterloo.ca (I.A.L.); melanie.maryanne.cousins@uwaterloo.ca (M.C.); smajowicz@uwaterloo.ca (S.E.M.); 4Global Studies Institute, University of Geneva, CH-1211 Geneva, Switzerland; anais.leger@blv.admin.ch (A.L.); didier.wernli@unige.ch (D.W.); 5Beijer Institute of Ecological Economics, Royal Swedish Academy of Sciences, 114 18 Stockholm, Sweden; 6WorldFish, Bayan Lepas 11960, Malaysia; 7Centre for Foodborne, Environmental and Zoonotic Infectious Diseases, Public Health Agency of Canada, Guelph, ON N1H 7M7, Canada; carolee.carson@canada.ca; 8Department of Population Medicine, Ontario Veterinary College, University of Guelph, Guelph, ON N1G 2W1, Canada; jparmley@uoguelph.ca

**Keywords:** antimicrobial resistance, antibiotic resistance, resilience, success factors, interventions, stewardship, public health, global health

## Abstract

Antimicrobial resistance (AMR) affects the environment, and animal and human health. Institutions worldwide have applied various measures, some of which have reduced antimicrobial use and AMR. However, little is known about factors influencing the success of AMR interventions. To address this gap, we engaged health professionals, designers, and implementers of AMR interventions in an exploratory study to learn about their experience and factors that challenged or facilitated interventions and the context in which interventions were implemented. Based on participant input, our thematic analysis identified behaviour; institutional governance and management; and sharing and enhancing information as key factors influencing success. Important sub-themes included: correct behaviour reinforcement, financial resources, training, assessment, and awareness of AMR. Overall, interventions were located in high-income countries, the human sector, and were publicly funded and implemented. In these contexts, behaviour patterns strongly influenced success, yet are often underrated or overlooked when designing AMR interventions. Improving our understanding of what contributes to successful interventions would allow for better designs of policies that are tailored to specific contexts. Exploratory approaches can provide encouraging results in complex challenges, as made evident in our study. Remaining challenges include more engagement in this type of study by professionals and characterisation of themes that influence intervention outcomes by context.

## 1. Introduction

Antimicrobial effectiveness has been diminishing due to microorganisms acquiring resistance [1]. Lagging efforts to invest in development of new antimicrobials [2] are further hampering effective treatment of resistant infections, contributing to thousands to millions of deaths every year [3,4,5] and to higher associated healthcare costs [6,7]. Antimicrobial resistance (AMR) is a global social-ecological challenge threatening public health and sustainable development in what is called a “One Health” issue [8].

Governments and institutions have taken actions to address AMR and minimise its consequences [3,9], implementing heterogeneous interventions targeting different settings and system levels with varying impacts. There is a need to address AMR from a “One Health” perspective and to consider social and ecological factors that drive AMR to better mitigate the problem. Effective actions to combat AMR could benefit from being imbedded within the broader agenda to build resilience in health systems and global public health, which will be critical for facing large-scale outbreaks and epidemics [10].

Interventions to tackle AMR can play a key role in building resilience towards AMR [11], but resources and information on interventions remain limited. Knowledge gaps include incompleteness of intervention design, implementation, and assessment [12]; and a lack of standardised reporting [11]. Assessment of interventions is often difficult and/or reported with a time delay. Hence, authorities often develop and implement interventions guided by biased information, implicit assumptions, and/or instinct or common beliefs, rather than structured theories that can be tested and evaluated [13]. AMR management is a dynamic challenge and, considering the importance of antimicrobials in modern medicine, there is a need to strengthen the science behind interventions against AMR [11]. While there are several studies assessing the effectiveness of interventions on AMR, currently, there also is a lack of translatable information about the effectiveness of interventions in different contexts because factors that make AMR interventions successful within and across a range of contexts are still poorly known [14].

While building a strong evidence base by incorporating implementation science in the design of interventions will take time, it is desirable to explore what information can be extracted from those that have already implemented interventions. Identifying factors that challenge or contribute to intervention success was a key objective of the present exploratory study, where success can be briefly defined as the intended goal and what each intervention wants to achieve. Based on our previous identification of interventions on AMR [11,15,16], we contacted the corresponding authors to answer a short survey asking for basic context, success factors, and obstacles about the intervention. We used thematic analysis to capture important themes that relate to positive intervention outcomes [17]. Studying success factors is important to understand what must be promoted—and under what circumstances—to have a positive impact on AMR or antimicrobial use (AMU) and could be essential to help learn, facilitate, and build resilience in future actions.

## 2. Results

Twenty-one interventions were analysed and their general contexts are shown in Table 1 (Appendix A [18,19,20,21,22,23,24,25,26,27,28,29,30,31,32,33,34,35,36,37,38]). The majority of interventions were perceived to have positive outcomes in reducing AMU and AMR levels and, thus, were in line with the main triggers of the interventions. Unintended consequences (i.e., outcomes that were not foreseen previously) involved higher consumption of other drugs, such as drugs for suppressing symptoms or narrow-spectrum or non-targeted antimicrobials, the latter reported also as a negative outcome because it contributed to increased resistance for them afterwards. However, the increased use of specific antibiotics was not enough to consider the intervention unsuccessful if overall antimicrobial use decreased.

Based on thematic analysis of responses, we found eight main success factor themes (Figure 1). Key sub-themes reported in at least one third of the interventions are detailed in Table 2. From the most frequent to the least, the first theme was behaviour of individuals or institutions towards the intervention. This theme encompasses the reinforcement of correct behaviour, trust/support, collaborative behaviour and flexibility, adaptability, and willingness to change towards the intervention or its implementation. Second, the theme institution characteristics covered the management with training, collaboration, and co-ordination; and the governance, including communication, partnership, and engagement/commitment. Third, information—available for, or resulting from, the intervention—with awareness, regulations/guidelines, outcomes, and data provision/collection was considered important. Fourth, intervention characteristics comprised factors such as multifaceted/multisector interventions, sustained in time (ongoing interventions or without an end date), scalability (possibility of implementing it top-down/bottom-up), and assessment of the intervention. Fifth, the capacity of the system where the intervention takes place was considered important. Within this theme, financial resources, including funding and financed training sub-themes, knowledge/skills of main actors, adequate premises and equipment, and novelty/innovation, were important sub-themes. The sixth theme, implementation characteristics of the intervention, included factors such as well-designed or well-planned, good, and detailed design and implementation, easy to implement, consultation or guidance during the implementation, and implementation supported by main actors. The seventh theme was AMU with the accessibility category, including sub-themes such as prescription, controlled, or monitored use or high cost; plus other sub-themes, including use reduction and use improvement. The last theme was infection control related to or resulting from the intervention and comprised surveillance, decrease in infection incidence, and prevention—hand hygiene/sanitation/vaccination. The 12 identified key sub-themes, which are those mentioned in at least one third of the 21 analysed interventions, were qualitatively mapped in Figure 2 to initially grasp what factors contributing to positive outcomes could be promoted together.

## 3. Discussion

Overall actions to maximise the chance of success in AMR interventions start long before their implementation and, once implemented, interventions still need to be flexible and supported with training, feedback, or advice. Behaviour and acceptance towards the intervention of individuals and institutions are often overlooked, but they seem to have a remarkable impact on success. Influencing behavioural change in combatting AMR is a challenge that involves not only the health sector, but many actors where we lack previous experience or who have not previously been engaged [39]. This is especially challenging when it has been emphasized that increasing AMR awareness does not necessarily transform into behavioural change or into logical changes in AMU practices simply because people in many cases do not choose their behaviour or it is constrained by broader factors [40,41]. This exemplifies the role of political, social, economic, and historical factors in AMU decision making, which are often not considered in our simplistic interpretations of knowledge-driven practices [40].

It seems that causes leading to successful interventions rely on reinforcement of new routines and correct behaviour that can be promoted by initial training and awareness campaigns, regular follow-ups with feedback, and refresher training that enhances continuous collaboration, motivation, and education. Attitudes and feelings were highlighted as key sub-themes (reinforcement of correct behaviour or new routines, trust and support, and collaboration) and these tend to induce motivation, security, and self-esteem of individuals, plus support and engagement from institutions. Confidence and awareness are established through engagement and shared responsibilities, with collaboration and co-ordination encouraging support of each other, humility, and reflexivity. These factors may lead to building trust and support between individuals, different role positions, and institutions, engaging support and collaboration between sectors and/or professionals. Behavioural change, awareness, and recognition of trade-offs associated with our actions have been highlighted cornerstones in the fight against AMR [39,42,43]. Moreover, well-established behaviours, such as routines and human tendency towards traditions and old habits, are not fully considered when designing or implementing interventions. This aligns with observations about guidelines, training, or even new and individualised policies being ineffective alone in achieving or maintaining behavioural change or to reform the industry sector [44,45,46].

Antimicrobial stewardship initiatives have been calling for the need to adopt a wider multidisciplinary and multifaceted approach involving experts from social, psychological, and behavioural sciences [45,47]. Assessment of stewardship initiatives in healthcare settings suggest that targets, actions, and locations are well-described but there is a lack of reporting about detailed actors (who is involved) and timing (how often and when), interfering with replicability and long-term behavioural change [48]. Although it is difficult to state how long support is needed for the intervention to be permanent or widely adopted, our findings suggest that impacting behaviour change starts in the design and implementation phases. All the above-mentioned information and the specific context and factors that can influence the behaviour of interest, both from individuals and from the system (socially, economically, and politically), need to be described [11,16] to avoid the tendency to return to or to fall on old routines once support, guidance, or feedback ends. There are internal dynamics to overcome following our and previous findings about addressing human behaviour as a cornerstone for successful intervention outcomes [48,49], which need to consider the whole context in which behaviour is embedded [11,15,44].

Themes related to AMU (AMU) and resistance prevalence (infection control) were the least frequently reported for having successful outcomes and, when they were, it was mainly as indicators (e.g., measures of AMU, AMR surveillance, or infection incidence). Resistance prevalence initiated a quarter of interventions, mainly focused on surveillance, which is still inconsistently implemented globally [50]; improving AMU initiated half of the interventions, mainly via stewardship programs. However, although AMU interventions are well-performed, outcomes may not be the ones expected or wanted if there is not deep knowledge of the complexity of the whole system (e.g., unexpected high resistance indicator after a strict stewardship program suggesting other sources of selective pressure [51]).

Evidently, financial resources are vital for success of interventions. Increasing budgets may not be easy but can benefit from public awareness (another key sub-theme), which can lead to institutional pressure to invest more at all levels. Most factors mentioned as obstacles to success (e.g., inertia and old involuntary habits) could be transformed into success factors (e.g., reinforcing new routines with continuous training, follow-ups, and/or feedback). Still, asking about obstacles and success factors separately can help identify features that contribute to success that may be overlooked or unnoticed when they work well or are taken for granted. Only one sub-theme (resources of the system), which belonged to the broader “capacity of the system” theme, was seen only as an obstacle (Appendix A). Nonetheless, this sub-theme probably is related to shortage of funding, which again emerged as an obstacle for success, and thus in line with other studies that have associated risky behaviours and actions that contribute to the spread of AMR with low economic income and poverty [39,40].

The information extracted highlights the value and importance of participatory studies to better address AMR. Differences between success factors and obstacles depending on context were not observed, probably as a consequence of the homogeneity of interventions analysed and the broad focus of the study. We also found that interventions were triggered reactively, largely focused on humans, and thus action was taken after a concern had arisen. However, AMR affects the “One Health” spectrum, which requires preventive, proactive, and broader interventions applied in multiple sectors to allow development of economies towards the Sustainable Development Goals that have direct or indirect relationships with AMR [52].

Better understanding of factors that contribute to intervention success would benefit from more qualitative research. While there are different qualitative approaches, thematic analysis has been widely accepted as a flexible and consistent framework for capturing perspectives [53]. Research about complex issues that are difficult to assess and to quantify, such as AMR, can advantageously make use of this type of analysis to help guide development of good practices before translatable information and assessments are available or accessible to other colleagues or when information is time-delayed. Publications often provided limited data on success factors and obstacles to intervention success. Because there is a need for fast publication and for confiding expertise of interventions to other colleagues, the main advantage of this analysis is that it adds value, complements, and improves new insights in interventional science learning from other previous experiences and their socioecological context. Unanticipated insights can enable and facilitate identification and selection of interventions and reduce the theory–practice gap, as well as strengthen the science of intervention against AMR using dissemination and implementation frameworks and making valuable information accessible to other colleagues [54,55].

Limitations of our study are connected to the fairly homogenous context, type, and location where the majority of analysed interventions took place and the broad aim of the study. Interventions are impacted by cultural, political, historical, and societal circumstances and publications do not follow reporting guidelines and, if they do, those may need to be updated for capturing relevant details to AMR. Moreover, types of interventions included were not equally represented nor equally assessed. All these facts can lead to restriction in capturing important themes to intervention success (e.g., identified success factors seem to be applicable in high-income countries but information and important themes affecting success of interventions in low-middle-income countries may still be missing).

Success factors mentioned in reports or publications are often more objective and easier to assess as they are in line with the intended effect of the intervention, while factors retrieved from this qualitative analysis reflected and relied-on subjective responses independent of the level of assessment of the intervention. Engagement of interdisciplinary experts, health professionals, designers, and implementers of interventions worldwide is needed to learn from non-traditional studies in AMR, such as participatory studies, and to capture key features of intervention success. Moreover, engagement of such profiles will help to have a more inclusive view and supporting information to capture or strengthen new themes or sub-themes that this analysis may have missed or may have identified as anecdotes. The unexpected consequences of our actions are difficult to foresee, but preparedness and system thinking approaches, as well as greater and more consistent reporting of experiences will equip interventionists with useful tools and knowledge before implementing new interventions. Broad system integration has been shown to positively affect resilience and tolerance in adverse situations or shocks [56]. Interventions success is also impacted by different baselines that are influenced by cultural, political, historical, and societal circumstances. Thus, better reporting in AMR is needed and initiatives proposed by Léger et al. [15] and Wernli et al. [11] can improve and align information to capture heterogeneity, even if small, of contexts that can affect AMR.

## 4. Materials and Methods

We examined interventions to gain a preliminary understanding as part of the AMResilience project [11,15,16], which compiled interventions identified through scientific databases and extracted the socioecological context of interventions using the AMR-Intervene framework [15], and, also, we added to that framework the respondent profile to identify which type of professionals engaged more in our exploratory study. We invited authors of interventions from different sectors to complete an exploratory questionnaire to identify additional relevant information and to describe facilitators and obstacles to their intervention success. The online questionnaire used open-ended questions to avoid delimiting or biasing their experiences. Then, following the standard for reporting qualitative research [57] (Appendix A), we performed an inductive thematic analysis to capture themes that relate to positive outcomes from interventions tackling AMR [17,57]. Coding was performed using MAXQDA v.2020, a computer-assisted qualitative data analysis software, without a pre-existing coding frame, and, therefore, was inductive, allowing the data to drive the themes (data-driven). To prevent bias from the main researcher, whose experience involves clinical microbiology and epidemiology, three more co-authors independently coded responses to assess interpersonal consistency. A theme was defined as the main explicit and clear tacit idea behind the participant answer and could be broken into sub-themes, which were detailed specific factors related to the main theme. A category was a (optional) middle classification where sub-themes could be related in a smaller group. Frequency was counted as interventions mentioning a particular theme, category, or sub-theme independently of the number of times repeated on each data item. Redundancies were included to not miss information, but, if present in the same data item, they were counted as one. Factors seen as key components for positive outcomes (either because factors were seen as satisfactory, or the opposite or lack of obstructive) were reorganised and clustered together (total frequency), although each contribution was considered in partial frequency. The method is described elsewhere [17] and, in our study, is also detailed in Appendix A. Finally, combining sub-themes and not relying on a single one may positively impact AMR, which emphasises the importance of needing to use several approaches to maximise success. Therefore, identified key sub-themes perceived as contributing to positive outcomes that were mentioned in the same response were qualitatively connected with the aim to initially grasp what could be promoted together.

## 5. Conclusions

Non-traditional analyses in the AMR field, such as exploratory qualitative studies, can provide us with novel evidence-based insights to alleviate the AMR problem from different perspectives in a proactive manner. Due to gaps in communication, reporting, and differences in contexts, replication of AMR interventions is difficult, with details and experiences often missing or elusive. Lessons from previous experiences can help to identify key factors for such interventions to be successful, and implementation frameworks will enhance interventional science and build resilience to AMR. Themes and sub-themes captured in our exploratory study are meant to encourage more complete appraisal, communication, discussion, and sharing of experiences among different professionals and, ultimately, help to inform context-dependant intervention design and implementation. The themes identified can also be useful for guiding production of reports best suited to helping policy development, which is the focus of many organisations and governments worldwide.

## Figures and Tables

**Figure 1 antibiotics-11-00639-f001:**
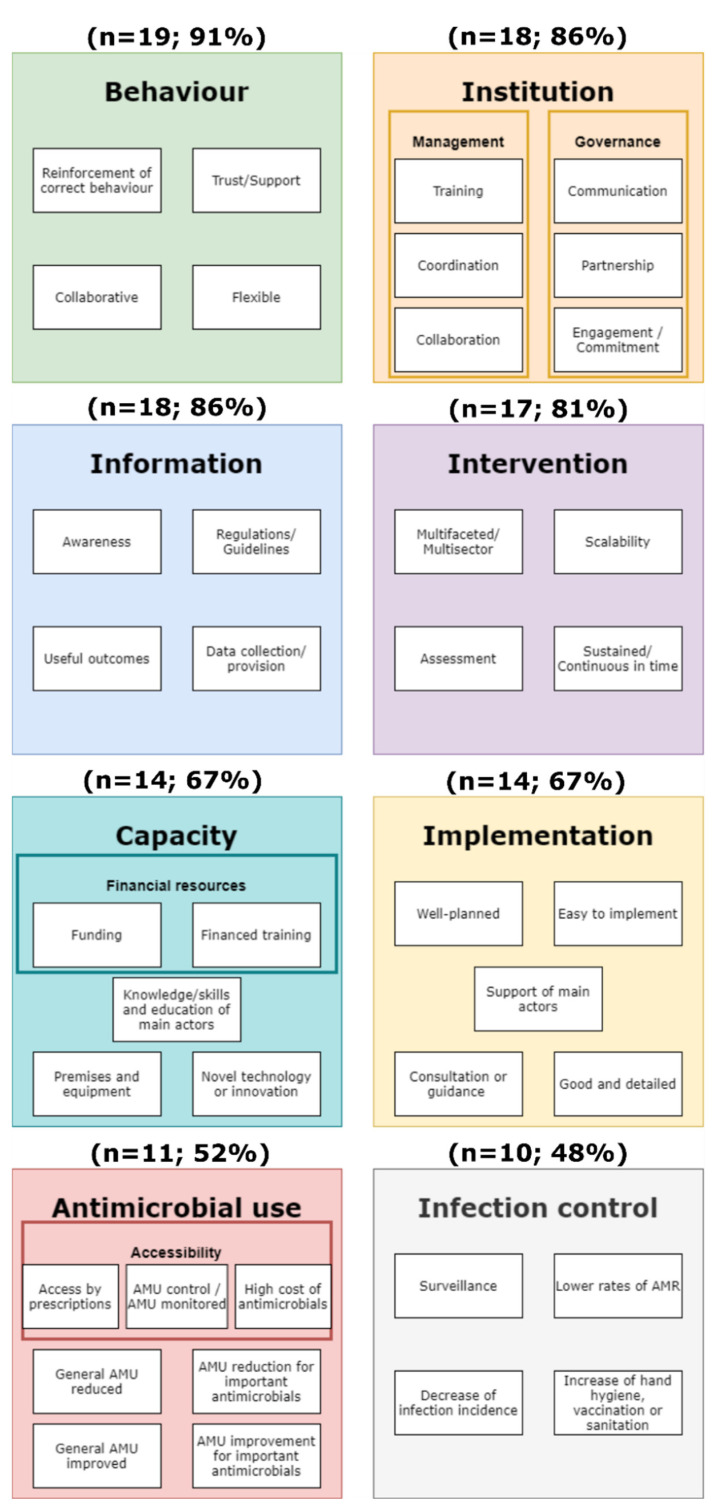
Key themes, categories, and sub-themes contributing to intervention success. Frequency and percentage of interventions reporting a particular theme are specified at the top of each block. Main blocks represent themes in different colour. Small white boxes represent sub-themes and, if present, categories are the coloured outline boxes grouping sub-themes. AMR = antimicrobial resistance; AMU = antimicrobial use.

**Figure 2 antibiotics-11-00639-f002:**
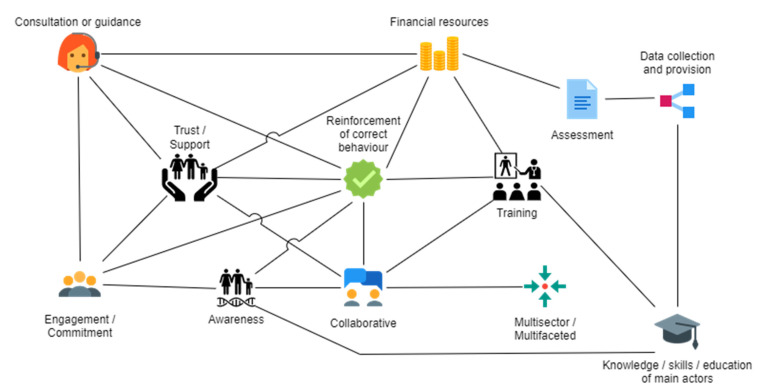
Qualitative map between the 12 most frequent sub-themes in our study. Connections between sub-themes were mapped when those were mentioned together in the same response, which illustrates factors that can be promoted together to enhance success or to positively impact AMR.

**Table 1 antibiotics-11-00639-t001:** Background information extracted from the 21 interventions using the AMR-Intervene framework [15]. * Component not inside of the AMR-Intervene framework [15]. AMR = antimicrobial resistance; AMS = antimicrobial stewardship; AMU = antimicrobial use; CA = Canada; MDR = multidrug resistance; N = number of interventions; OH = “One Health”; SDR = simple drug resistance; US = United States.

Component [15]	Variables [15]	Categories			N		Percentage	
**Respondent ***	**Profile**	**Medical doctor and University Professor**			7		33%	
		Researcher			6		29%	
		Public Health Epidemiologist			4		19%	
		University Professor (Pharmacist/Veterinarian)			2		10%	
		Medical doctor			2		10%	
Publication	Year	Before 2010			5		24%	
		2010–2014			8		38%	
		2015–2019			8		38%	
	Quality of description	Sufficient	(Detailed)		14	5	67%	24%
			(Good)			5		24%
			(Succinct)			4		19%
		Vague			7		33%	
Social system	Economic scale	High-income countries			20		95%	
		Low-middle-income countries (Nepal)			1		5%	
	Spatial scale	Europe			13		62%	
		Asia (Japan, Israel, and Nepal)			4		19%	
		North America (US and CA)			3		14%	
		Australia			1		5%	
	Sector scale	Human sector			14		67%	
		Animal sector			3		14%	
		Human, Animal, Food sectors (“OH”)			3		14%	
		Animal and Food sectors			1		5%	
	Time scale	Start		Before 2010	14		67%	
				2010–2014	5		24%	
				2015–2019	2		10%	
		End		Delimited	9		43%	
				Ongoing/without end	12		57%	
Governance	Agents responsible	Public sector (government owned)			17		81%	
		Private sector (private owned)			1		5%	
		Academic sector (research/scientific sector)			1		5%	
		Public and Private sector			1		5%	
		Public and academic sector			1		5%	
	Level of funding	Specific funding		Public funding	11	10	52%	48%
				Private funding		1		5%
		Without funding			10		48%	
	Constitution	Positive: initiate or improve			18		86%	
		Negative: refrain or prevent			3		14%	
	Policy instrument	Information/awareness			14		67%	
		Regulations			4		19%	
		Information/awareness and regulations			3		14%	
Trigger/goals	Trigger of the intervention	Pressure on AMR (high AMU)			6		29%	
		State of AMR (increase of AMR)			4		19%	
		Driver of AMR			3		14%	
		Impact of AMR (mortality/morbidity)			3		14%	
		Pressure and impact of AMR			3		14%	
		Pressure and state of AMR			2		10%	
	Trigger type	Reactive			12		57%	
		Preventive			9		43%	
	Main goal towards intervention	Initiate an action			15		71%	
		Improve an action			5		24%	
		Maintain an action			1		5%	
	Main strategy	Reduce AMU			12		57%	
		Surveillance			6		29%	
		Infection prevention			2		10%	
		Innovation			1		5%	
	System intervention	Low leverage point			20		95%	
		High leverage point			1		5%	
	Level of implementation	National			13		62%	
		Local			5		24%	
		Sub-national or Regional			3		14%	
Bio-ecological scale	Type of microorganism	Bacteria			16		76%	
		No specific			4		19%	
		Fungi			1		5%	
	Level of resistance	Unknown			12		57%	
		MDR			8		38%	
		SDR			1		5%	
	Resistance coming from plasmids	Unknown or not specified			15		71%	
		Yes			5		24%	
		No			1		5%	
	Host carrier of AMR microorganism	Human			14		67%	
		Animal			4		19%	
		Human, animal, and/or food			3		14%	
	Main transmission of resistance	Human to human			10		48%	
		Unknown/Not specified			7		33%	
		Animal or Food to human			2		10%	
		Human, animal, or food to human			1		5%	
		Environment or human to human			1		5%	
	Origin of infection	Not applicable			11		52%	
		Healthcare associated			6		29%	
		Community acquired			3		14%	
		Healthcare or community acquired			1		5%	
	Climate	Temperate			19		91%	
		Arid/Temperate			2		10%	
Assessment	Cost-effectiveness	Not evaluated			17		81%	
		Evaluated			4		19%	
	Main outcome	Pressure: Reduction in AMU			10		48%	
		State: Reduction in AMR prevalence			6		29%	
		Impact: Less morbidity and mortality			3		14%	
		Drivers: Improvement in sanitation			2		10%	
	Outcome(s) perception or evaluation	Positive			16		76%	
		Positive and negative			4		19%	
		Not reported/neutral			1		5%	
	Unintended outcomes	Not reported			14		67%	
		Reported			7		33%	

**Table 2 antibiotics-11-00639-t002:** Meaning, assumptions, implications, and supporting quotes of the 12 key sub-themes, which were reported in at least one third of the interventions, resulting from the thematic analysis. The main theme where they belong and the total and partial frequency of each are also specified. AMR = antimicrobial resistance, AMU = antimicrobial use; FREQ = frequency; GPs = General Practitioners (medical doctors); IC = infection control; OB = obstacle; PREV = prevalence; SF = success factor. * Category (sub-themes: financed training and funding counted together as they are extremely related).

Sub-Theme	Theme	Total FREQ	Partial FREQ		Meaning and Assumptions	Implications	Quotes
			SF	OB			
Reinforcement of correct behaviour, new mentality, or changes	Behaviour	10	6	4	New routines or ideas must be reinforced to ensure their continuity. Habituation needs time and going back to old routines due to inertia is usual.	Training and guidance are essential to make changes in the long term. Follow-up and regular feedback maintain motivation. Sustained efforts and sustained interventions are needed to avoid old habits. Use of new technologies (emoticons or mass media) as reminders.	“change in mentality should also be seen as one of the key success factors of this study” // “There is a human tendency to return to previous practices in the absence of constant motivation and reminders” // “Inertia among prescribers”
Financial resources *	Capacity	10	5	5	Enough budget and funding to carry out the intervention. Funding for teaching and training the main actors responsible for the intervention.	Good level of funding is crucial for implementing interventions. Costs can be very high including training, personnel, or resources and, without a proper budget, many of them are not going forward.	“Coaching of farmers” // “clinically oriented education through symposia, workshops and focused meetings at the regional and local levels” // “funding” // “budget to begin with” or “very costly to establish”
Assessment	Intervention	9	9	0	After the intervention has taken place or, for a defined period of the ongoing intervention, checking or measuring outcomes of the actions applied can help to elucidate the usefulness of the intervention or its possible gaps.	Results from assessment can help to maintain motivation and to identify new goals and opportunities to improve outcomes or to promote actions impacting AMR.	“results from the monitoring were used when writing guidelines” // “a decline of 26.5% in the number of antibiotic prescriptions was observed over 5 years” // “significantly increased the usage of hand-rub dispensers in patient rooms, in comparison to the three other tested conditions”
Awareness	Information	9	6	3	Knowledge about AMR and people aware of the problem of untreatable infections enhance positive outcomes. Ignorance of the problem may lead to opposition of public opinion or citizenship (e.g., lack of prescription thought as cutback in health system)	Society may behave differently following and finishing prescribed antimicrobial treatments. Prescribers less pressured to prescribe treatments to please patients or farmers. Citizenship engage to preserve antimicrobial effectiveness.	“Patients who insisted on receiving antibiotics” // “Health beliefs by the general populations” // “general reluctance amongst farmers and veterinarians to change their existing antimicrobial treatment practices” // “advisor/coach helps the farmer with explaining what he/she could be improving and what the risk is when certain practices are not performed correctly”
Knowledge, skills and education	Capacity	9	9	0	Deep and detailed knowledge or education increases system capacity to carry out the intervention.	Contribution and expansion of skills and knowledge to new staff or new performers of interventions. Impede waning of the intervention.	“investment in technical and epidemiological knowledge” // “The programme created a pool of trained technicians who can compensate for transfer and separation of staff and contribute to expansion of programme staff”// “advisor/coach helps the farmer with explaining what he/she could be improving and what the risk is when certain practices are not performed correctly”
Trust and support	Behaviour	8	4	4	Trust and support of main actors. On the contrary, prejudices and scepticism hamper good outcomes.	They enhance implementation and maintenance of interventions.	“supported by the key doctors of the ICU” // “Sustained efforts and trust of infectious disease pharmacists” // “Perception of the farmer that interventions cost money and time (although often not the case as proven in our studies)
Training	Institution	8	8	0	Professional training of the actions to improve or initiate in the intervention.	Training empower and increase self-esteem to carry out interventions, especially when actors are not familiar on a daily basis with AMR. Often, this training is funded.	“Training of GPs”// “The programme created a pool of trained technicians who can compensate for transfer and separation of staff and contribute to expansion of programme staff” // “farmers keep control over the health situation and are less reluctant to change certain AMU treatment procedures” // “pump priming investment to support development of pharmacists”
Multifaceted/Multisector	Intervention	8	8	0	Intervention is composed or carried out by different sectors, settings, departments, or professionals	Interventions not only affect one type of actors or sectors. Joining efforts from different backgrounds and perspectives may have bigger impacts, reach, and redundancy of interventions. Some tasks can be carried out or complemented by different agents for completion	“Involving community pharmacists, care homes, nursing homes staff in this process and using training and care pathways” // “Intensive collaboration between the surveillance team and the medical microbiologists” // “close collaboration between the animal and human sector and between experts and political stakeholders or authorities”
Consultation and guidance	Implementation	8	7	1	Consultation or guidance for intervention implementation clarify actions and objectives of the intervention. Consultation and guidance available for actors. When lacking, often implies insecurity towards the intervention and actors can go back to old habits	This tool during implementation or during the intervention enhances positive outcomes as they can rely on experts or other professionals’ criteria when doubts arise. It promotes self-esteem and motivation of executors due to continuous knowledge, feedback and follow-ups	“development of practical implementation guidance” // “advice from the Expert Advisory Group on Antimicrobial Resistance ceased in 2004” // “to achieve this (AMU) reduction, it is important to assist and guide farmers in this process”
Collaborative	Behaviour	7	4	3	Collaboration between main actors enhances implementation and communication. In contrast, reluctance to participate due to fear of consequences that may not reflect reality hinder implementation	A collaborative behaviour is crucial to involve individuals into the fight of AMR, especially those coming from private sectors. Popular beliefs and ignorance can jeopardise the designed intervention	“It came from industry and therefore was well adopted” // “There still exist hesitance among slaughterhouses to participate due to the fear of losing customers, if resistant bacteria are found” // “recruitment of herds was difficult, despite the efforts made to promote this study and the possibility for farmers to collaborate free of charge”
Engagement/support	Institution	7	6	1	Compromise towards the intervention not only from individuals but also from the institutions designing, implementing, or performing the intervention	Ensures effort from the institution to maintain or to carry out the intervention, independently of individual governances.	“Implementation had the support of heads of both departments” or “the veterinarian has often already been the advisor for years resulting in the loss of motivation due to, for example persistent disease problems”
Data collection and provision	Information	7	5	2	Data collection and provision standardised, available, and shared. In contrast, data collected or provided from different sites, with heterogeneous criteria or not shared hinders availability of knowledge	Data of interventions that are shared, with standard reporting, can clarify the exact situation of the epidemiological state; these can be used by different settings or sectors and can clarify and/or quantify assessments.	“This integrated program was made possible because access to all relevant data and samples that were already systematically collected from animals, food, and humans has been shared” // “number of tools to make surveillance findings transparent and accessible to both scientists and non-specialists” // “Diversity in coding in laboratory information systems”

## Data Availability

De-identified data are available on request. Data requests with an expression of interest in pursuing secondary or extended analyses with a specific research question can be made to the corresponding author.

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
