# Peer review of "Studying Factors Affecting Success of Antimicrobial Resistance Interventions through the Lens of Experience: A Thematic Analysis"

_antibiotics, 2022, doi:10.3390/antibiotics11050639_

Round 1

Reviewer 1 Report

I congratulate the authors for a very interesting paper. Qualitative research is sometimes not easy to apprehend for quantitative researchers like me, despite the clear recognition of its role in implementation science and as a complementary tool to enrichen findings by avoiding the a priori conditioning by the researchers’ previous knowledge. In this paper, the authors used the right tool and were successful in conveying a message from their data.

I have some suggestions mainly related with presentation issues/options.

Lines 33-34 (abstract): The sentence “More engagement by professionals and characterization by context of themes shaping intervention outcomes are still remaining challenges.” is difficult to follow, please make it syntactically simpler to be clearer.

The first sentence of the Introduction (lines 41-42) would be better placed at the end of the following paragraph, after current references 7,8, in current line 46.

The structure within section RESULTS doesn’t work well. Please avoid the subsection title “2.1. Figures and Tables”. Figures and Tables are embedded in the main text and adequately cited in the text of the Results section. That is enough.

Lines 98-100: “Therefore, when the twelve identified key sub-themes perceived as contributing to positive outcomes were mentioned in the same response they were qualitatively connected in Figure 2 to initially grasp what could be promoted together.” This should be Methods. In the results describe minimally what is presented in Figure 2.

Lines 100-102: “Differences between success factors and obstacles depending on context were not observed, probably as a consequence of the homogeneity of interventions analysed and the broad focus of the study.” This sounds as Discussion to me.

The title of Table 2 (“Most frequent sub-themes and themes related to the success of interventions resulting from the thematic analysis”) is unspecific. This almost could be the legend of Figure 1… Please be more informative about the contents of this table. Also, it would improve readability a lot if you used less abbreviations, particularly for themes. If necessary, insert this table in landscape oriented pages.

The legend of Figure 2 is insufficient. Without going to the text, it is not clear what “qualitative connections” are. Also, the spatial display of elements does not facilitate communication of a message. Using straight lines does not favour you since there is one connection with no solution (the upper line linking Engagement and Awareness) and a lot of lines crossing. The geometric nature of your representation induces the reader to try to find some meaning in being in line (either horizontally or vertically). In summary, I suggest something more like a network analysis representation for this result.

The methods section is too short. Part of the content of Table S2 and its reference (1. Braun, V.; Clarke, V. Using thematic analysis in psychology. Qual. Res. Psychol. 2006, 3, 77–101.) should appear in the main text.

It is not clear how many participants from the team of each intervention answered the questionnaire. The results presentation suggests that the “intervention” is the unit of observation, as if there were one questionnaire answer for each intervention. But then how did the authors guarantee there were “health professionals, designers and implementers of AMR interventions” answering the questionnaire?

Line 213-214: “inductive thematic analysis to capture evidence-based themes that relate to positive outcomes from interventions tackling AMR” – “inductive” and “evidence-based” sound contradictory to me. As the authors themselves explain well, inductive refers to results emerging from the respondents’ opinion, which the authors tried not to be conditioned by their own previous knowledge.

Line 212: “following the standard for reporting quality research [36] (Table S1, Supplementary Material)” should read “qualitative” instead of “quality”.

The reference under Table S1 is reference 36 of the main text, but this is not made clear and the format should be the same to facilitate readability.

The publications in Table S3 should be provided with their full reference.

Author Response

Reviewer 1:

I congratulate the authors for a very interesting paper. Qualitative research is sometimes not easy to apprehend for quantitative researchers like me, despite the clear recognition of its role in implementation science and as a complementary tool to enrichen findings by avoiding the a priori conditioning by the researchers’ previous knowledge. In this paper, the authors used the right tool and were successful in conveying a message from their data.

I have some suggestions mainly related with presentation issues/options.

Lines 33-34 (abstract): The sentence “More engagement by professionals and characterization by context of themes shaping intervention outcomes are still remaining challenges.” is difficult to follow, please make it syntactically simpler to be clearer.

Authors’ response: Thank you for this comment. Changes have been made to make the sentence clearer.

The first sentence of the Introduction (lines 41-42) would be better placed at the end of the following paragraph, after current references 7,8, in current line 46.

Authors’ response: Thank you for this suggestion. Suggested change has been applied.

The structure within section RESULTS doesn’t work well. Please avoid the subsection title “2.1. Figures and Tables”. Figures and Tables are embedded in the main text and adequately cited in the text of the Results section. That is enough.

Authors’ response: Thank you for this suggestion. Suggested change has been applied.

Lines 98-100: “Therefore, when the twelve identified key sub-themes perceived as contributing to positive outcomes were mentioned in the same response they were qualitatively connected in Figure 2 to initially grasp what could be promoted together.” This should be Methods. In the results describe minimally what is presented in Figure 2.

Authors’ response: Thank you for this comment. Changes have been applied inserting that sentence in Methods and describing better Figure 2 in Results as follows: “Finally, combining sub-themes, and not relying on a single one, may positively impact AMR which emphasizes the importance of needing to use several approaches to maximize success. Therefore, identified key sub-themes perceived as contributing to positive outcomes that were mentioned in the same response were qualitatively connected with the aim to initially grasp what could be promoted together.” and “Figure 2: Qualitative map between the twelve most frequent sub-themes in our study. Connections between sub-themes were mapped when those were mentioned together in the same response which illustrates factors that can be promoted together to enhance success or to positively impact AMR”, respectively.

Lines 100-102: “Differences between success factors and obstacles depending on context were not observed, probably as a consequence of the homogeneity of interventions analysed and the broad focus of the study.” This sounds as Discussion to me.

Authors’ response: Thank you for this suggestion. The sentence has been place in Discussion (“The information extracted highlights the value and importance of participatory studies to better address AMR. Differences between success factors and obstacles depending on context were not observed, probably as a consequence of the homogeneity of interventions analysed and the broad focus of the study. We also found that…”.

The title of Table 2 (“Most frequent sub-themes and themes related to the success of interventions resulting from the thematic analysis”) is unspecific. This almost could be the legend of Figure 1… Please be more informative about the contents of this table. Also, it would improve readability a lot if you used less abbreviations, particularly for themes. If necessary, insert this table in landscape oriented pages.

Authors’ response: Thank you for this comment. We have reduced the abbreviations in the table to make it more clear and readable and we have detailed better the header of Table 2 (“Table 2:  Meaning, assumptions, implications and supporting quotes of the twelve key sub-themes, which were reported at least in one third of the interventions, resulting from the thematic analysis. The main theme where they belong and the total and partial frequency of each are also specified. AMR= antimicrobial resistance, AMU= Antimicrobial use; FREQ= Frequency; GPs= General Practitioners (medical doctors); IC= Infection Control; OB= obstacle; PREV= prevalence; SF= success factor. *category (sub-themes: financed training and funding counted together as they are extremely related).”).

The legend of Figure 2 is insufficient. Without going to the text, it is not clear what “qualitative connections” are. Also, the spatial display of elements does not facilitate communication of a message. Using straight lines does not favour you since there is one connection with no solution (the upper line linking Engagement and Awareness) and a lot of lines crossing. The geometric nature of your representation induces the reader to try to find some meaning in being in line (either horizontally or vertically). In summary, I suggest something more like a network analysis representation for this result.

Authors’ response: Thank you for this comment. We have explained better Figure 2 in the legend (“Figure 2: Qualitative map between the twelve most frequent sub-themes in our study. Connections between sub-themes were mapped when those were mentioned together in the same response which illustrates factors that can be promoted together to enhance success or to positively impact AMR.”) and, also, we have updated Figure 2 to make it more readable and understandable.

The methods section is too short. Part of the content of Table S2 and its reference (1. Braun, V.; Clarke, V. Using thematic analysis in psychology. Qual. Res. Psychol. 2006, 3, 77–101.) should appear in the main text.

Authors’ response: Thank you for this comment. We have now referenced the article by Braun and Clarke, which is in the introduction and methods section with reference number 17, and we have extended the methods section including information from the supplementary Table S2, which has also the reference and the reference number of the main text.

It is not clear how many participants from the team of each intervention answered the questionnaire. The results presentation suggests that the “intervention” is the unit of observation, as if there were one questionnaire answer for each intervention. But then how did the authors guarantee there were “health professionals, designers and implementers of AMR interventions” answering the questionnaire?

Authors’ response: Thank you for this comment. Indeed, we got an answer from each of the interventions which accounts for 21 participants. Therefore, to know who was responding the questionnaire we added to the AMRIntervene framework used in Table 1 the respondent profile. The respondent profile is not included in the mentioned framework and with that we wanted also to inform which profiles engaged in our study, which typically are the profiles that help guide, design and implement interventions. The profiles covered the following areas: Medical doctor and University Professor (7); Researcher (6); Public Health Epidemiologist (4); University Professor (Pharmacist/Veterinarian) (2) and Medical Doctor (2).

Line 213-214: “inductive thematic analysis to capture evidence-based themes that relate to positive outcomes from interventions tackling AMR” – “inductive” and “evidence-based” sound contradictory to me. As the authors themselves explain well, inductive refers to results emerging from the respondents’ opinion, which the authors tried not to be conditioned by their own previous knowledge.

Authors’ response: Thank you for this comment. The term evidence-based has been deleted to avoid confusion. What we wanted to say is that these themes are based on the experience and perception of professionals with their respective intervention.

Line 212: “following the standard for reporting quality research [36] (Table S1, Supplementary Material)” should read “qualitative” instead of “quality”.

Authors’ response: Thank you for this comment. This change has been applied.

The reference under Table S1 is reference 36 of the main text, but this is not made clear and the format should be the same to facilitate readability.

Authors’ response: Thank you for this comment. We have formatted the references and we have also specified the reference number corresponding to the main manuscript in Table S1 and also for Table S2.

The publications in Table S3 should be provided with their full reference.

Authors’ response: Thank you for this comment. We have provided the full reference for each article.

Reviewer 2 Report

The authors are presenting a description of ‘themes’ that were reported to be associated with antibiotic prescribing decisions.  Their methods are not necessarily clear, but presume this was a questionnaire applied retrospecitively.

Authors could consider this specific improvements regarding the methodology: More description of their actual methods (questionnaire, oral survey, etc.).  The reported 21 response but how many individuals contributed? 

I thought charts/tables were confusing.and didn’t present things in a concise manner.  I suspect it is because the data is from a small subset and lacking a bit in the material presented.  Overall, I had a hard time reading the paper and found my mind wandering.  I think it lacks scientific merit but if you want to publish a ‘descriptive study’ then it would be suitable.

Please reference AMU (line 68) in your text.

Author Response

Reviewer 2:

The authors are presenting a description of ‘themes’ that were reported to be associated with antibiotic prescribing decisions.  Their methods are not necessarily clear, but presume this was a questionnaire applied retrospecitively.

Authors’ response: Thank you for this comment. Methodology has been expanded and explained more in detail.

Authors could consider this specific improvements regarding the methodology: More description of their actual methods (questionnaire, oral survey, etc.).  The reported 21 response but how many individuals contributed?

Authors’ response: Thank you for this comment. We have detailed this in the Introduction and, also in the Methods section. Indeed, we got an answer from each of the interventions which accounts for 21 participants. Therefore, to know who was responding the questionnaire we added to the AMRIntervene framework used in Table 1 the Respondent Profile, which is not included in the mentioned framework, with the aim to picture better which profiles engaged more.

I thought charts/tables were confusing.and didn’t present things in a concise manner.  I suspect it is because the data is from a small subset and lacking a bit in the material presented.  Overall, I had a hard time reading the paper and found my mind wandering.  I think it lacks scientific merit but if you want to publish a ‘descriptive study’ then it would be suitable.

Authors’ response: Thank you for this comment. We have changed and explained better figures and tables, we have reduced abbreviations in Table 2 and we have updated Figure 2.

Please reference AMU (line 68) in your text.

Authors’ response: Thank you for pointing this. It has been referenced.

Reviewer 3 Report

This is an article to establish the preliminary interventions identified through scientific databases and the socio-ecological context of interventions in the factors affecting success of antimicrobial resistance interventions. It reflects some general interventions in antimicrobial resistance which is a global health problem and it focus in the other aspects that are very important in this topic and they are less visible: communication, partnership and engagement/commitment, the trust/ support with collaborative behavior and flexibility, adaptability and willingness to change towards the intervention or its implementation. The authors mentioned the multifaceted/multisector interventions to get the aim of changing the antimicrobial resistance and antimicrobial use.

The manuscript is clear, relevant for the field and presented with well-structured manner. The highlighting areas of the article is the importance of the topic and the scientific soundness. The authors get the objective to give general rules to success in changing antimicrobial resistance. Figure 1 (the eight main success factor themes) is very interesting because it represents the frequency and percentage of interventions reporting every particular theme in the articles evaluated. We notice that the most important themes are behavior, institution, information and intervention, in contrary with the antimicrobial use and infection control that are less discussed in the articles when whe want to success in global antimicrobial resistance interventions. The conclusions are consistent with the evidence and arguments presented.

Author Response

Reviewer 3:

This is an article to establish the preliminary interventions identified through scientific databases and the socio-ecological context of interventions in the factors affecting success of antimicrobial resistance interventions. It reflects some general interventions in antimicrobial resistance which is a global health problem and it focus in the other aspects that are very important in this topic and they are less visible: communication, partnership and engagement/commitment, the trust/ support with collaborative behavior and flexibility, adaptability and willingness to change towards the intervention or its implementation. The authors mentioned the multifaceted/multisector interventions to get the aim of changing the antimicrobial resistance and antimicrobial use.

The manuscript is clear, relevant for the field and presented with well-structured manner. The highlighting areas of the article is the importance of the topic and the scientific soundness. The authors get the objective to give general rules to success in changing antimicrobial resistance. Figure 1 (the eight main success factor themes) is very interesting because it represents the frequency and percentage of interventions reporting every particular theme in the articles evaluated. We notice that the most important themes are behavior, institution, information and intervention, in contrary with the antimicrobial use and infection control that are less discussed in the articles when whe want to success in global antimicrobial resistance interventions. The conclusions are consistent with the evidence and arguments presented.

Authors’ response: Thank you for your feedback.

Reviewer 4 Report

Dear Authors,

I started reading with interest your manuscript, because the topic is highly relevant, as you mention. 

Trying to know the causes of the success of different interventions sounds quite promising to design future studies and interventions.

On the other hand, thinking about the One Health approach also sounds quite interesting.

Notwithstanding this, I have different concerns after reading the manuscript and, in my opinion, the following point limits the value of the proposed study.

Namely:

(1) It is not clear to me how the included studies were selected. In my opinion, this is crucial to be able to repeat the study, if necessary.

(2) The manuscript focused on the "success" of interventions. Notwithstanding this, after reading the manuscript I have not been able to locate the definition of success.

(3) Taking into account the small number of studies included in the analysis, I'm not sure if including non-human studies is a good decision in this case, because human-health studies and animal-health studies are quite different, and have different designs, and "success" is different.

Author Response

Reviewer 4:

Dear Authors,

I started reading with interest your manuscript, because the topic is highly relevant, as you mention. 

Trying to know the causes of the success of different interventions sounds quite promising to design future studies and interventions.

On the other hand, thinking about the One Health approach also sounds quite interesting.

Notwithstanding this, I have different concerns after reading the manuscript and, in my opinion, the following point limits the value of the proposed study.

Namely:

(1) It is not clear to me how the included studies were selected. In my opinion, this is crucial to be able to repeat the study, if necessary.

Authors’ response: Thank you for this comment. We have introduced the following sentence in the introduction to make this clearer in the manuscript (“Based on our previous identification of interventions on AMR [11,15,16], we contacted the corresponding authors to answer a short survey asking for basic context, success factors and obstacles about the intervention.”) and details are stated in Methods (“We examined interventions to gain a preliminary understanding as part of the AMResilience project [11,15,16], which compiled interventions identified through scientific databases and extracted the socio-ecological context of interventions using the AMR-Intervene framework [15] and, also, we added to that framework the respondent profile to identify which type of professionals engaged more in our exploratory study . We invited authors of interventions from different sectors to complete an exploratory questionnaire to identify additional relevant information and to describe facilitators and obstacles to their intervention success. The online questionnaire used open-ended questions to avoid delimiting or biasing their experiences.”)

 (2) The manuscript focused on the "success" of interventions. Notwithstanding this, after reading the manuscript I have not been able to locate the definition of success.

Authors’ response: Thank you for this comment. We have detailed the following sentence to make this clear in the introduction and now it reads as follows: (“Identifying factors that challenge or contribute to intervention success was a key objective of the present exploratory study, where success can be briefly defined as the intended goal and what each intervention wants to achieve.”). We have also changed this sentence in the discussion, and now it reads as follows: (“Success factors mentioned in reports or publications are often more objective and easier to assess as they are in line with the intended effect of the intervention, while factors retrieved from this qualitative analysis reflected and relied on subjective responses independent of the level of assessment of the intervention.”

(3) Taking into account the small number of studies included in the analysis, I'm not sure if including non-human studies is a good decision in this case, because human-health studies and animal-health studies are quite different, and have different designs, and "success" is different.

Authors’ response: Thank you for this comment. Although it is true that the majority of the studies focus on human studies, we have not seen any differences with the factors to be promoted if we compare human and animal interventions in this study. Therefore, taking animal interventions out will reduce the richness of the study while the results and conclusions will remain the same. We are not defining success for every intervention, as interventions are tailored depending on social-ecological context, and therefore the meaning of success in each intervention may vary. However, the factors to arrive to the desired outcome seem to be similar independently of type of intervention and context of the intervention in our exploratory study.

Round 2

Reviewer 4 Report

Dear Authors,

Thank you for your answer, and provide a revised version of the manuscript.

In my opinion, the detailed description of the methods has greatly improved the quality of the study and the interpretation of the results.

My only suggestion, at this point, is that the manuscript could improve if a paragraph describing the limitations of the design, the results, and a general sentence on the limitations of the included studies, which have an impact on the overall study.

Kind regards

Author Response

Dear Authors,

Thank you for your answer, and provide a revised version of the manuscript.

In my opinion, the detailed description of the methods has greatly improved the quality of the study and the interpretation of the results.

My only suggestion, at this point, is that the manuscript could improve if a paragraph describing the limitations of the design, the results, and a general sentence on the limitations of the included studies, which have an impact on the overall study.

Authors’ response: Thank you for this comment. We have updated the manuscript with the following paragraph in the “Discussion”:

“Limitations of our study are connected to the fairly homogenous context, type, and location where the majority of analysed interventions took place and the broad aim of the study. Interventions are impacted by cultural, political, historical and societal circumstances and publications do not follow reporting guidelines and, if they do, those may need to be updated for capturing relevant details to AMR. Moreover, types of interventions included were not equally represented nor equally assessed. All these facts can lead to restriction in capturing important themes to intervention success (e.g.: identified success factors seem to be applicable in high-income countries but information and important themes affecting success of interventions in low-middle-income countries may still be missing).”